# Ethnographic meta-analysis shows that thermoregulation activities predict needle and awl use in North America

**McKenna Lynn Litynski**[1]*, **Sean Field**[2], **Randall Haas**[1]

**1** Department of Anthropology, University of Wyoming, Laramie, Wyoming, United States of America,
**2** School of Computing and Department of Anthropology, University of Wyoming, Laramie, Wyoming, United States of America

* mlitynsk@uwyo.edu

## Abstract

Needles and awls are common artifacts in the perishable archaeological record. To understand behavioral motivations for their use, this study examines ethnographic activities linked to needles and awls in North America. We hypothesize that thermoregulation would have been the strongest driver of perforator use. Ethnographies from eHRAF World Cultures are examined to evaluate activity types and temperature effects on the prevalence of perforator use. We observe that the sum of non-thermoregulation activities (69%) account for the majority of ethnographic perforator occurrences. Such activities include tattooing, medical suturing, basketry, and ceremonial activities among others. Nonetheless, the most prevalent activity identified is clothing production, accounting for 14% of ethnographic observations. We furthermore observe from a series of linear mixed-effect models that account for spatial autocorrelation that the probability of perforator observations increases at the coldest temperatures and decreases at the warmest, with a 52% probability of observing perforator tools ethnographically at −35.5°C and a 37% probability of observing perforators where temperatures reach +12.9°C. These results support the hypothesis that thermoregulation, particularly clothing manufacture, was a major driver of perforator tool use while simultaneously revealing that such tools were also commonly deployed in a wide array of activities. Such findings provide insight into the environmental and socio-cultural factors that influenced the use of perforators and thus inform our understanding of an artifact class that is increasingly observed in the archaeological record.

## Introduction

Perforator tools made from bone, including *needles* and *awls* (Fig 1), are not uncommonly observed in Paleolithic perishable assemblages around the world [1,2]. Needles are perforator objects that consist of a long groove and a point/tip and are meant

**Data availability statement:** R code associated with this manuscript can be found at this github link: https://github.com/mlitynsk/NA_bonetools Currently, this github repository is now available for public viewing. The GitHub repository includes the updated linear mixed-effect models. All other relevant data are within the paper and its Supporting Information files.

**Funding:** The author(s) received no specific funding for this work.

**Competing interests:** The authors have declared that no competing interests exist.

to be passed through a material, while awls increase in width away from the tip but do not have an eye or groove and are not passed completely through the medium [3,4]. Such tools offer critical insight into how early human populations solved the adaptive challenges of Late Pleistocene and Early Holocene global expansion. Both tool types have been used by humans for thousands of years, with bone awls dating to as early as ~75 ka [5–7] and needles dating to at least 45 ka [5]. At first glance, perforators might seem to be tools for producing and maintaining leather clothing and housing, which allowed Paleolithic populations to enter and survive cold Pleistocene environments [5,8–11]. However, ethnographic observations suggest that such tools also served a variety of other purposes unrelated to thermoregulation. For example, perforators are commonly used in the production of basketry, tattooing, and wood working [12–19]. The extent to which perforator tools can be confidently linked to thermoregulatory activities or other purposes remains unclear, limiting our understanding of how influential these tools were in shaping past human migration into cold environments.

This analysis approaches the problem by compiling ethnographic accounts of North American needle and awl use from the eHRAF World Cultures database. We are interested in determining the extent to which needles and awls were created and used for manufacturing thermoregulation items or alternative purposes. Under a thermoregulation framework, cold climates necessarily compel humans to invest in thermoregulatory technologies that often include perforator tools. This model makes two basic empirically testable predictions for the ethnographic record. First, we should expect that perforator tools are most often associated with thermoregulation activities in the ethnographic record. Second, it anticipates that environmental temperatures predict the use of perforator tools. The frequency of perforator tool observations in the ethnographic record ought to be greater in cold climates than in warm climates. Such predictions align with Osborn's [11] observation that a 1,300-year cold snap between 12.9–11.6 cal. ka known as the Younger Dryas Cold Event significantly increased the importance of thermoregulatory technology such as the manufacture of tailored skin clothing and thus inflated the production of needles for sewing.

Although there is good reason to suspect that thermoregulation was the major driver of perforator technology, qualitative observation of ethnographic literature shows that needles and awls commonly serve a variety of functional and symbolic purposes outside of the realm of thermoregulation. Examples of such needle and awl functions include tattooing human skin, beadwork, fishing, basketry, woodwork, and serving as gifts for ceremonies [3,12,14–18]. When used in these capacities, needles and awls are not limited to cold environments.

Moreover, while previous research shows that perforators figure prominently in the production of cold-weather clothing, at least one prominent counter example shows that such technologies are not requisite for cold-climate adaptation. Namely, Garvey [19] highlights how ethnographic Patagonian populations that inhabited cold polar environments did not make tailored leather clothing nor is there evidence of needles, awls, or microblades in the Late Pleistocene archaeological deposits of Patagonia. The lack of evidence for bone perforator tools suggests individuals may have had

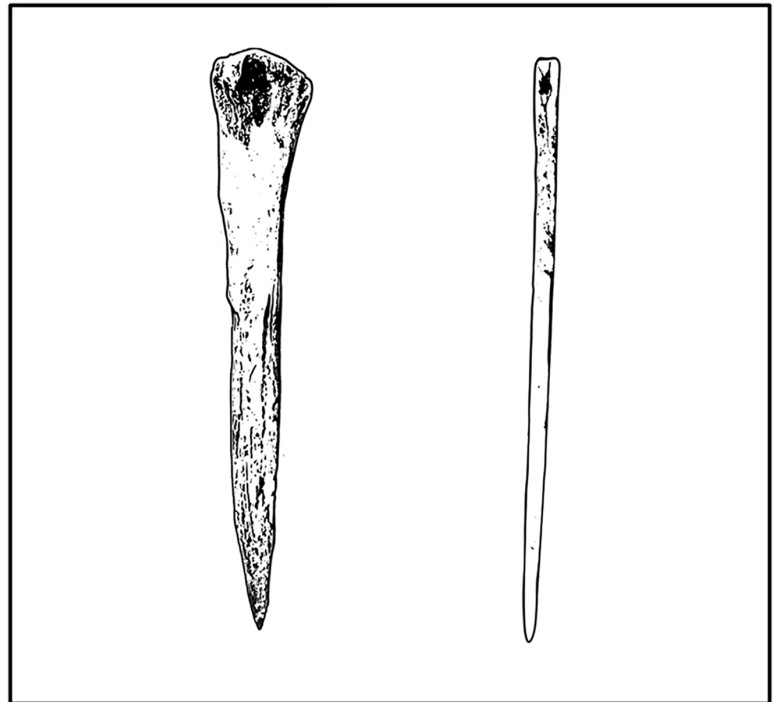

**Fig 1. Illustration of bone sewing needles (right) and awls (left). Artwork produced by Brenna Litynski with permission for publication.**

alternative physiological and behavioral adaptations to the cold, and the colonization of this region during the Late Pleistocene was likely slow and thus the technology and knowledge to manufacture fitted clothing was lost [19].

These varied examples highlight that while perforator tools are linked to thermoregulation activities, the extent to which this is true remains unclear. Such tools likely serve as multi-tools for any combination of activities. The two models of perforator use represent lenses through which we can understand the extent to which needles and awls were used in cold stress adaptations or other applications.

At one extreme, we could suppose that environmental temperatures exerted a strong effect on the use of such tools. At the other extreme, we could suppose that some other combination of perforator functions overwhelms any climatic effects. The answer has implications for our interpretation of bone perforators in the archaeological record and thus our understanding of human behavior more generally, especially global migrations. The purpose of this meta-analysis is to determine the magnitude of climatic and other effects on the use of perforator tools among ethnographic societies.

## Materials and methods

This study evaluates two predictions derived from the working model. First, the model anticipates that the majority of ethnographic observations of perforator tools (including needles and awls) will primarily be associated with thermoregulatory activities including clothing, shelter, bedding, and blanket manufacture and maintenance. Second, the model anticipates that the probability of observing the use of perforator tools will increase in colder environments. Ethnographic observations of perforator use in North America are used to evaluate these predictions. We select North America as a case study for understanding perforator tool use based on the ability to obtain data from Indigenous groups spanning extremes of cold and warm conditions at a continental scale. Additionally, the ethnographic observations, unlike archaeological observations, afford a view of the perishable toolkits and behavioral adaptations associated with perforator tools. Here we

focus on an ethnographic approach, remaining alert to its analytical limitations and with an eye toward a complimentary archaeological approach in the future. This section describes creating the perforator tool database, operationalization of environmental temperature data, and analytical methods of evaluating tool use types and environmental temperature relationships.

## Creating the perforator tool database

The ethnographies compiled to test these predictions are from the eHRAF World Cultures (electronic Human Relations Area Files) database. All eHRAF data include paragraph citations presented in S1 Table. The data presented from the eHRAF World Cultures database are published under a CC BY license, with permission granted from eHRAF. The eHRAF database provides access to ethnographic texts [20] from over 300 Indigenous and ethnic groups located in a variety of regions associated with all possible historical climate zones across North America.

We first search all eHRAF records for the terms, "needle" and "awl" (S1 Table). For each record mentioning needles or awls, the tool type, publication citation, page number, activity type (e.g., clothing, blankets, tattooing, basketry, mats, etc.), activity classification (e.g., thermoregulatory or alternative), and perforator material are recorded. Activity types related to the creation or maintenance of materials likely to be used for thermoregulatory purposes (e.g., blankets, clothing, shelter, bedding, mats, etc.) are classified as "thermoregulatory" and all activities related to materials not directly related to thermoregulation (e.g., basketry, ceremonies, tattooing, storytelling, fishing, etc.) are classified as "alternative". If a given tool is associated with multiple activities by the same eHRAF source, then each activity is recorded as a separate observation.

We remain alert to several limitations and biases that come with deriving data from eHRAF World Cultures. First, we recognize the impact of ethnocentrism and how certain activities associated with needles and awls may not be included in the ethnographic documents. In this light, we also recognize the inherent gender bias that can skew the perspectives, experiences, and information shared in the ethnographic record [21–23]. As a first approximation, we have no theoretical reason to believe these biases compromise the integrity of our analysis because such biases ought to play out equally across the entire geographical and temperature domains of this study. At most, they may only diminish the statistical power of our analysis. We focus on qualitatively and quantitatively evaluating the ethnographic data and associated perforator activities directly recorded in eHRAF World Cultures. Additionally, we assess the relationship between perforator tool activity types and temperature by qualitatively evaluating geographical data by plotting our ethnographic database in space in addition to selecting specific search terms specific to perforator tools. The coded data and the temperature trends associated with perforator tool use remains relevant and useful to understanding the wide variety of uses for needles and awls.

## Environmental temperature data

Our analysis requires the assignment of a temperature variable to each ethnographic case. We use the average minimum temperature of the coldest month (MTCM) to capture thermoregulatory pressures that a given ethnographic group likely experienced. This proxy seems particularly salient relative to common alternative proxies such as effective temperature [24–26] because MTCM quantifies the extreme or limiting environmental conditions that impact people's ability to survive in cold environments and thus the dependence of clothing for human survival. Furthermore, MTCM most precisely captures the selective forces driving thermoregulation, with previous scholars noting that it is often extreme events that shape cultural selection rather than averages [27,28]. We obtain MTCM data from WorldClim version 2.1 [29], a database consisting of high-resolution weather and climate information between the years 1970 and 2000 on a global scale presented in geographic format (i.e., GeoTiff). The MTCM for each ethnographic group in the database is based on the geographic centroid of the group's territory based on eHRAF World Cultures, with MTCM data derived from bioclimatic variable BIO6 from WorldClim (Fig 2). This included groups associated with perforator tool technologies and those with no mention of

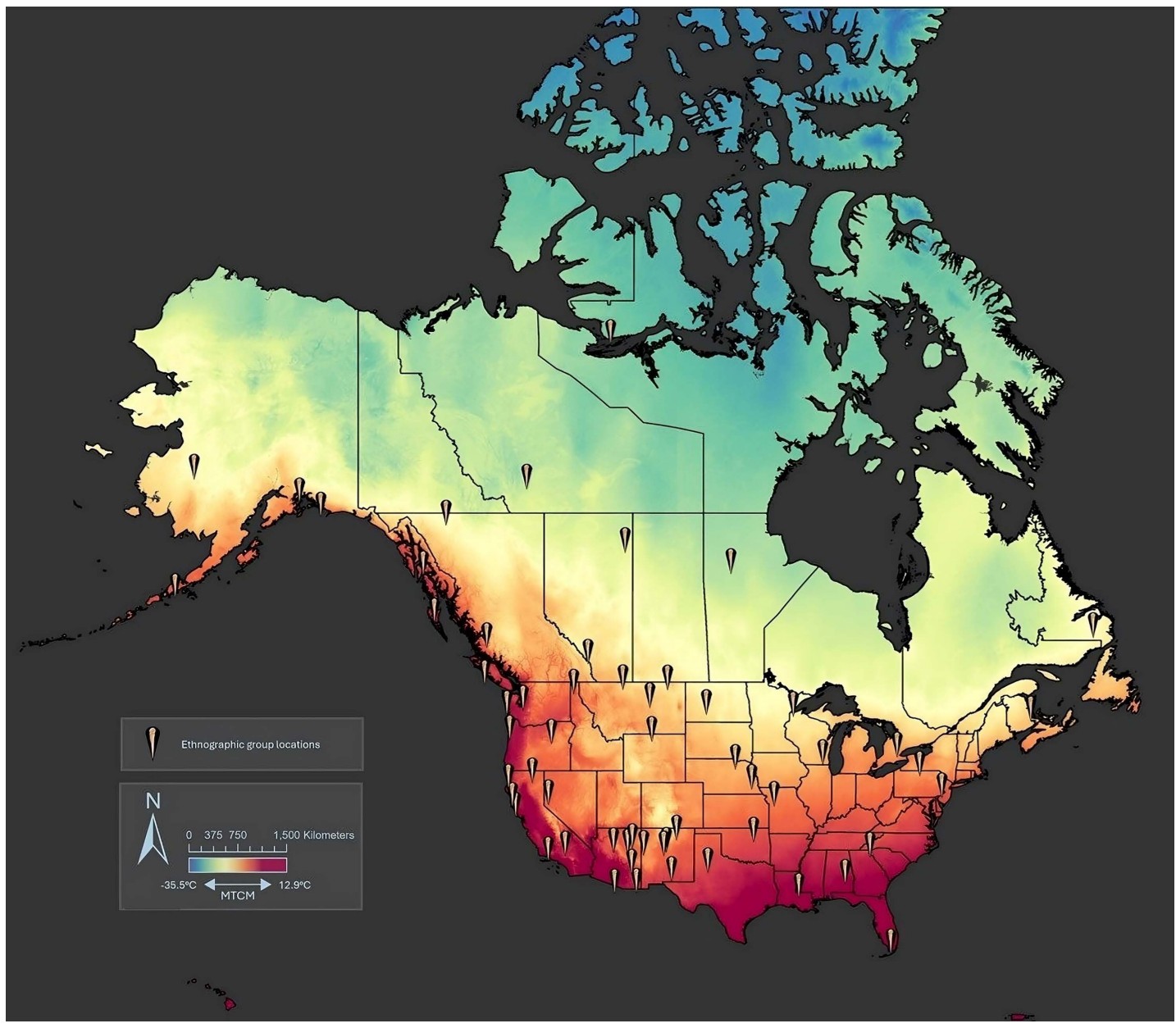

**Fig 2. Heat map featuring the average minimum temperature of the coldest month from WorldClim and points across North America depicting the geographical centroids of 59 Indigenous groups territories available in the eHRAF World Cultures database.** Latitude and longitude information republished from the eHRAF World Cultures database under a CC BY license, with permission from Dr. Carol Ember, original copyright 1993.

needles/awls ethnographically. Non-Indigenous cultural groups within eHRAF were not included in this study. All temperature extraction routines are performed by ArcGIS.

We recognize that the WorldClim climate data do not span the entire period of ethnographic data in this study spanning the 19th-21st centuries. However, temporal mismatches are on the order of 0–150 years. At this temporal scale and given that our dataset spans some 50 degrees C, any MTCM errors are certain to be trivial. Additionally, paleoclimatic datasets

are based on a series of interpolations and assumptions. Such assumptions would likely introduce greater error relative to WorldClim which provides high-resolution temperature data across all geographical regions in North America, ensuring consistent and comparable data between cultural groups. For this reason, WorldClim is the most appropriate dataset for the analysis and is most likely to minimize error in the temperature variable considered in this analysis.

## Statistical analysis

To test the prediction that the majority of perforator tools were used for thermoregulation, the total number of ethnographic documents within the "thermoregulation" and "alternative" categories were converted to percentages with the expectation that the proportion of thermoregulatory activities should exceed that of other activities. A chi-square test is used to evaluate statistical significance of observed differences in proportions.

To assess the effect of MTCM on perforator tool occurrences, we conduct a series of linear mixed-effect logistic regressions. The first model examines the effect of MTCM on the frequency of perforator occurrences per ethnographic record. The second model examines the effect of MTCM on the frequency of thermoregulation perforator activities per ethnographic record. The third model evaluates the effect of MTCM on the frequency of non-thermoregulatory perforator activities per ethnographic record. Additional models evaluate the effect of MTCM on the frequency of major activity types. For all models, we account for differential representation of documents by incorporating the total frequency of documents associated with each Indigenous group as a random-effect variable. We furthermore take into account the longstanding issue of spatial autocorrelation, also known as Galton's Problem [30] by including latitude and longitude coordinates of each ethnographic group as a random effect. We do this using the Matérn correlation function implemented with the spaMM R statistical computing environment [31,32]. Error terms are reported as predictive variance, which is used to denote the uncertainty in the linear predictor [31].

## Results

The eHRAF database includes ethnographic observations from 59 North American Indigenous cultural groups. Of these, needle use is observed in the ethnographies of 55 groups (93%) represented in 278 documents (S1 Table). Awl use is observed among 54 groups (92%) within 273 literature sources. Of the 467 individual documents examined, we record 1,191 separate needle or awl observations in the ethnographic record. This includes instances where multiple activities were associated with the same eHRAF document. Here we use these observations to assess the extent to which thermoregulatory and other activities anticipate the use of perforator tools in the ethnographic record of North America.

## Tool function

Two-hundred and twenty-three mentions of needles and/or awls in the eHRAF ethnographic record fall under the generic "Toolkit" category and thus cannot be assigned to a specific activity. Tools in this "Toolkit" category are nonetheless considered in statistical models evaluating the relationship between environmental temperature and ethnographically observed needles and awls. We record a total of 92 distinct activities. The most frequent activity type observed in association with perforator tools is "Clothing" with 133 observations (Table 1). The second most frequent activity is ceremonies or rituals with 76 observations. Other activities with more than 20 observations include basketry (n=61), tattooing (n=59), shoes (n=48), piercings (n=37), storytelling (n=35), trading (n=35), mats (n=33), shelter (n=31), medical (n=29), snowshoes (n=26), blankets (n=21), and fishing (n=20; Table 1; Fig 3). When considering the frequency of thermoregulation activities (n=301, 31%) and those activities associated with alternative purposes (n=665, 69%), we observe a statistically significant difference ($X^2$=137, $df$=1, $p$=<0.01). We therefore observe that the occurrence rate of alternative activities, in aggregate, is approximately double that of thermoregulatory activities. However, clothing manufacture and maintenance is the most prominent single activity associated with perforator tools. These observations indicate that while clothing production and thus thermoregulation do largely motivate the use of perforator tools, alternative activities are also important.

Table 1. Table featuring the various perforator tool uses, the number of occurrences within the ethnographic record unique to awls and needles individually, the total count of both needles and awls within the ethnographic record, and whether these uses are related to thermo-regulation or if these tools were used for alternative purposes. The table also provides a description of how each use type has been classified within the ethnographic record.

| Activity category | Thermo-regulation? | Total Tool (count) | Total Tool (%) | Awl (count) | Awl (%) | Needle (count) | Needle (%) |
|---|---|---|---|---|---|---|---|
| Toolkit | NA* | 223 | NA* | 136 | NA* | 87 | NA* |
| Clothing | Yes | 133 | 13.77% | 62 | 12.60% | 71 | 14.98% |
| Ceremony or Ritual | No | 76 | 7.87% | 41 | 8.33% | 35 | 7.38% |
| Baskets | No | 61 | 6.31% | 52 | 10.57% | 9 | 1.90% |
| Tattooing | No | 59 | 6.11% | 11 | 2.24% | 48 | 10.13% |
| Shoes | Yes | 48 | 4.97% | 29 | 5.89% | 19 | 4.01% |
| Piercing | No | 37 | 3.83% | 21 | 4.27% | 16 | 3.38% |
| Storytelling | No | 35 | 3.62% | 25 | 5.08% | 10 | 2.11% |
| Trading | No | 35 | 3.62% | 25 | 5.08% | 10 | 2.11% |
| Mats | Yes | 33 | 3.42% | 5 | 1.02% | 28 | 5.91% |
| Shelter | Yes | 31 | 3.21% | 16 | 3.25% | 15 | 3.16% |
| Medical | No | 29 | 3.00% | 1 | 0.20% | 28 | 5.91% |
| Snowshoes | Yes | 26 | 2.69% | 6 | 1.22% | 20 | 4.22% |
| Blankets | Yes | 21 | 2.17% | 6 | 1.22% | 15 | 3.16% |
| Fishing | No | 20 | 2.07% | 3 | 0.61% | 17 | 3.59% |
| Foodways | No | 16 | 1.66% | 11 | 2.24% | 5 | 1.05% |
| Beadwork | No | 15 | 1.55% | 2 | 0.41% | 13 | 2.74% |
| Bags | No | 13 | 1.35% | 6 | 1.22% | 7 | 1.48% |
| Boat Covering | No | 13 | 1.35% | 6 | 1.22% | 7 | 1.48% |
| Name | No | 13 | 1.35% | 13 | 2.64% | 0 | 0.00% |
| Gifts | No | 12 | 1.24% | 6 | 1.22% | 6 | 1.27% |
| Hunting | No | 12 | 1.24% | 2 | 0.41% | 10 | 2.11% |
| Magic | No | 12 | 1.24% | 2 | 0.41% | 10 | 2.11% |
| Punishment | No | 12 | 1.24% | 2 | 0.41% | 10 | 2.11% |
| Scratching | No | 12 | 1.24% | 0 | 0.00% | 12 | 2.53% |
| Gaming | No | 11 | 1.14% | 9 | 1.83% | 2 | 0.42% |
| Metalwork | No | 11 | 1.14% | 11 | 2.24% | 0 | 0.00% |
| Dance | No | 10 | 1.04% | 7 | 1.42% | 3 | 0.63% |
| Quillwork | No | 10 | 1.04% | 9 | 1.83% | 1 | 0.21% |
| Art | No | 9 | 0.93% | 9 | 1.83% | 0 | 0.00% |
| Other Non-Thermoregulatory | No | 132 | 13.66% | 92 | 18.70% | 40 | 8.44% |
| Other Thermoregulatory | Yes | 9 | 0.93% | 2 | 0.41% | 7 | 1.48% |

*NA is assigned to the "Toolkit" category based on the generalized nature of such accounts in the ethnographic record. It is not possible to assign "Tool-kit" counts to either thermoregulation or alternative use, and we did not include this use category in GLM models or percentage contributions considered in this paper.

We furthermore observe clear differences in the activity types performed with needles and awls, often but not always in obvious ways. For example, there are more observations of awls used for ceremonies/rituals, baskets, shoes, piercings, storytelling, and instances of trading compared to needles (Table 1). Needles are used more frequently for clothing manufacture, tattooing, mats, medical use, snowshoes, blankets, and fishing (Table 1). Both perforator tool types are represented in relatively equivalent proportions for manufacturing shelter from both plant and animal materials.

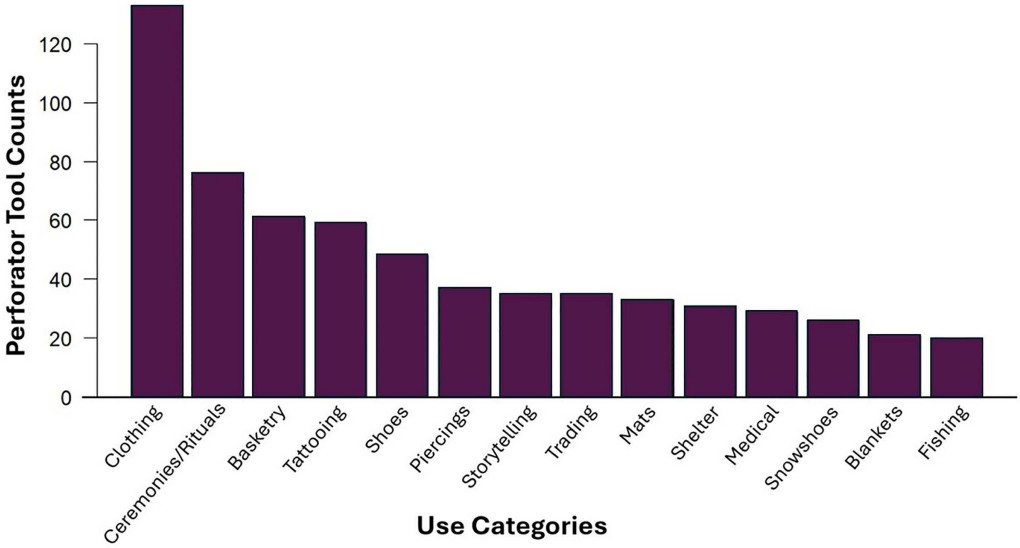

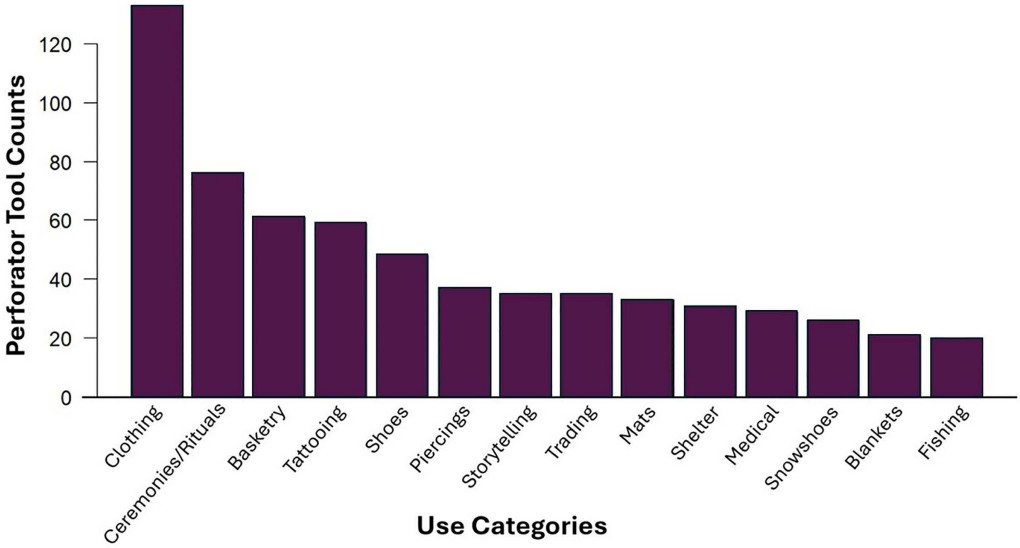

**Fig 3. Bar graph showcasing activities associated with needles and awls within the eHRAF World Cultures database along with the number of instances each activity was recorded as part of the meta-analysis.** Only those activities observed on 20 or more occasions in the ethnographic record are featured in this graph.

## Perforators and temperature

The MTCM temperatures for 59 cultural groups under consideration range between −35.5°C and +12.9°C. Statistical results indicate a negative relationship between the probability of observing perforator tools in the ethnographic record and MTCM ($p = 0.07$). The logistic regression results indicate that at an MTCM of −35.5°C there is a 52±15% probability of observing perforator tools ethnographically, dropping to a 37±14% probability at an MTCM of +12.9°C (Fig 4; Table 2). When considered separately, we once again observe a negative relationship between MTCM and ethnographically observed needles ($p = 0.08$) and awls ($p = 0.10$). There is a 35±25% chance of observing a bone needle among ethnographic groups living in regions where MTCM averages −35.5°C dropping to an 18±23% probability of observing needles among groups living where MTCM reaches +12.9°C. Similarly, in regions where MTCM averages −35.5°C, there is a 37±30% chance of observing an awl dropping to a 20±31% chance at +12.9°C. These results consistently show that lower temperatures predict increased likelihood of perforator use regardless of tool type but that temperature effects are stronger on needle use compared with awl use.

Logistic regression is also used to assess the effects MTCM on perforator tools classified as "thermoregulation" versus those classified as "alternative purposes." Not surprisingly, we discover a negative relationship between MTCM and the probability of observing needles and awls used specifically for thermoregulatory activities in the ethnographic record ($p = 0.05$). We find that probability of observing a perforator tool drops from 26±38% at an MTCM of −35.5°C to 9±40% among accounts at an MTCM of +12.9°C. In contrast, we do not detect a statistically significant relationship between the prevalence of perforator tools used for alternative purposes and temperature data ($p = 0.40$; Fig 4; Table 2).

Analysis of tools used for several other distinct activity types shows strong negative relationships between temperature and occurrences of needles or awls. These activities include the manufacture or use of clothing, footwear, snowshoes, and fishing related activities ($p < 0.01$, Table 2). Except for fishing, we note that these are all thermoregulatory activities. Other individual uses of needles and awls do not appear to be related to temperatures. Examples of such activities with no significant relationship between temperature and perforator tools occurrences include tattooing ($p = 0.30$), ceremonies or

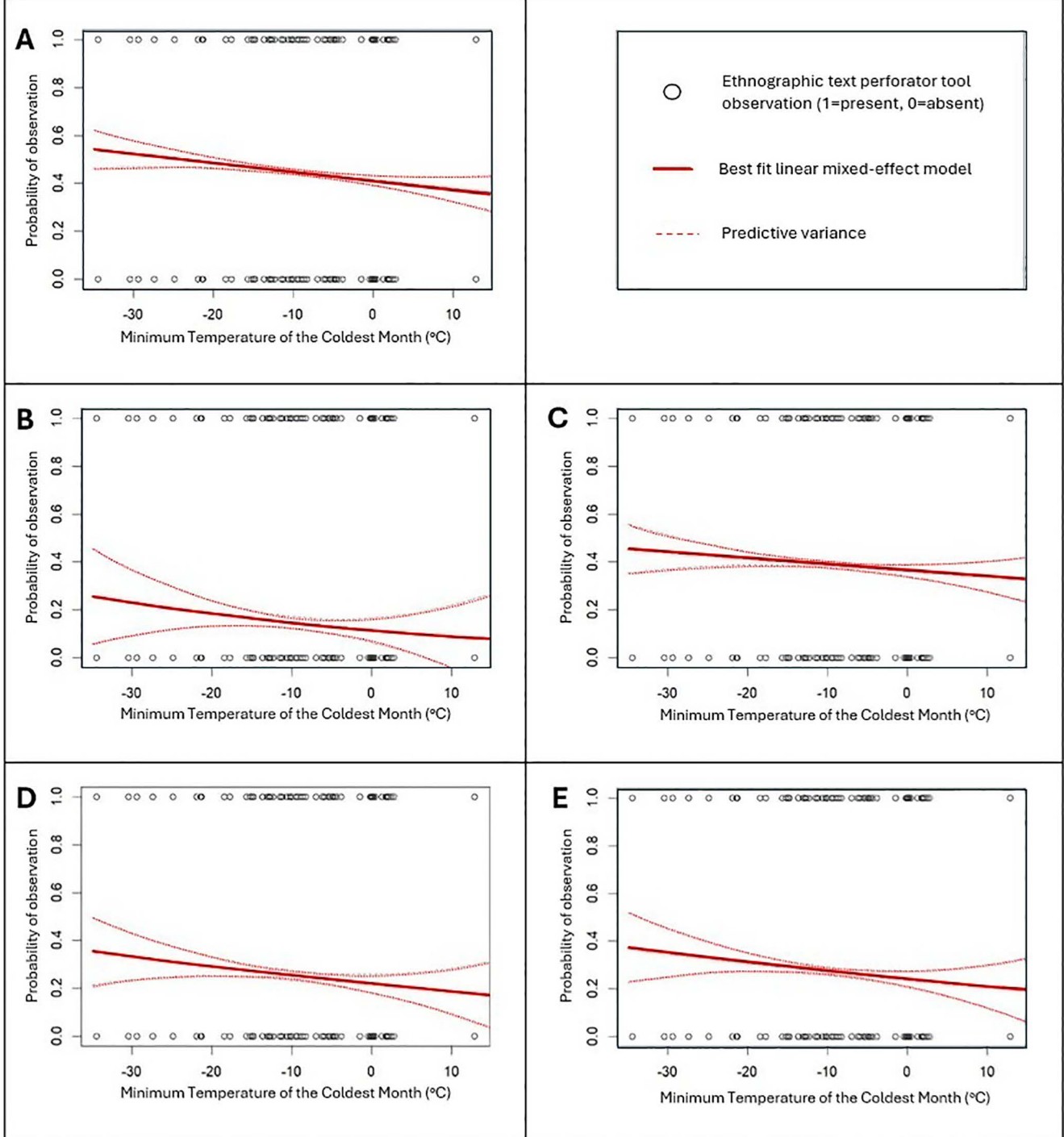

**Fig 4. Logistic regression results showing the probability of observing a perforator tool within the eHRAF ethnographic database as a function of the average temperature of the coldest month (°C) as observed in the WorldClim database.** Ethnographic observations are denoted by open black circles, with 1 indicating presence and 0 indicating absence. Red lines represent best fit logistic regressions and dotted lines show predictive variance. A. all perforator tools. B. thermoregulation observations. C. alternative observations. D. needles only. E. awls only.

**Table 2. Results of logistic regression models showing negative relationships between perforator use and environmental temperature primarily for thermoregulatory activities but not alternative activities.**

| Model | Estimate | Standard Error | T-Value | P-Value | Tool Occurrence Sample Size |
|---|---|---|---|---|---|
| Perforator Tool Count | −0.015 | 0.010 | −1.450 | 0.074** | n = 1,191 |
| Needle Count | −0.020 | 0.014 | −1.382 | 0.084** | n = 563 |
| Awl Count | −0.012 | 0.014 | −1.272 | 0.102* | n = 628 |
| Perforator Tool Count (Thermoregulation Use) | −0.028 | 0.016 | −1.694 | 0.046** | n = 301 |
| Perforator Tool Count (Alternative Use) | −0.003 | 0.012 | −0.248 | 0.401 | n = 665 |
| Perforator Tool Count (Tattooing Usage) | −0.015 | 0.027 | −0.536 | 0.296 | n = 59 |
| Perforator Tool Count (Basketry Usage) | 0.089 | 0.029 | 3.034 | 0.999 | n = 61 |
| Perforator Tool Count (Clothing Usage) | −0.030 | 0.016 | −1.869 | 0.031*** | n = 133 |
| Perforator Tool Count (Ceremony/Ritual Usage) | 0.004 | 0.021 | 0.208 | 0.582 | n = 76 |
| Perforator Tool Count (Piercing Usage) | −0.030 | 0.045 | −0.680 | 0.248 | n = 37 |
| Perforator Tool Count (Shoes Usage) | −0.046 | 0.027 | −1.720 | 0.043*** | n = 48 |
| =Perforator Tool Count (Storytelling Usage) | 0.014 | 0.116 | 0.123 | 0.549 | n = 35 |
| Perforator Tool Count (Mats Usage) | 0.091 | 0.057 | 1.605 | 0.946 | n = 353 |
| Perforator Tool Count (Shelter Usage) | −0.017 | 0.029 | −0.582 | 0.280 | n = 31 |
| Perforator Tool Count (Medical Usage) | −0.002 | 0.032 | 0.049 | 0.520 | n = 29 |
| Perforator Tool Count (Snowshoes Usage) | −0.163 | 0.067 | −2.415 | 0.008*** | n = 26 |
| Perforator Tool Count (Blankets Usage) | 0.020 | 0.035 | 0.586 | 0.721 | n = 21 |
| Perforator Tool Count (Fishing Usage) | −0.085 | 0.032 | −2.673 | 0.004*** | n = 20 |

*p-value is less than 0.1.

**p-value is less than 0.01.

***p-value is less than 0.05.

rituals ($p = 0.58$), piercing ($p = 0.25$), storytelling ($p = 0.55$), medical use ($p = 0.52$), shelter ($p = 0.28$), and blankets ($p = 0.72$), basketry ($p = 1$), and mats ($p = 0.95$; Table 2).

## Discussion

Ultimately motivated to advance our understanding of archaeological perforator tools, this meta-analysis began by asking whether environmental temperature influences the prevalence of perforator tool uses in the ethnographic record, and to what extent needles and awls are associated with thermoregulation activities compared with alternative uses. The working model anticipated that thermoregulation activities would be the primary driver of perforator tool use. We hypothesized that needles and awls would be observed more frequently among ethnographic cultures living in colder environments as opposed to warmer climates. These predictions were tested by examining perforator tool prevalence and use types within the eHRAF ethnographic record, and comparing those observations with environmental temperature data (WorldClim MTCM) associated with 59 Indigenous groups represented in the eHRAF World Cultures database. The representation of perforator tools among 100% of the groups considered in this study highlights the importance of this technology on a continental scale.

The results of the meta-analysis fail to support the initial prediction that the majority of occurrences of needles and awls in the eHRAF World Cultures database would be associated with thermoregulation. Rather, non-thermoregulatory activities significantly outnumber thermoregulatory activities indicating the importance of needles and awls for alternative functions, most notably ceremonies/rituals, basketry, tattooing, piercing, discussions of needles or awls in storytelling, trading, and fishing (S1 Table). Nonetheless, the manufacture of clothing—a distinctly thermoregulatory activity—is the single most prevalent activity associated with perforator tools with 133 ethnographic observations. Moreover, six out of the eight

thermoregulation activities associated with perforator tools occur in the top 20 coded categories, with four of those within the top 10 most frequently cited categories, underscoring the central importance of thermoregulation in motivating the use of perforator tools. We therefore find general support for the predicted association between thermoregulatory activities and perforator use while also observing a wide range of non-thermoregulatory activities.

The data furthermore support the second prediction derived from the working model that the probability of observing perforator tools in the eHRAF World Cultures ethnographic record would be negatively related to temperature. This hypothesis holds true when all perforator tools in aggregate, when only considering needles and awls, and when considering those tools used specifically for thermoregulation. As expected, this trend is not observed when considering the frequencies of perforator tools used for alternative purposes with an exception among the variables we tested including fishing, which also shows a negative relationship with temperature.

The finding of a clear relationship between temperature and perforator use in the ethnographic record reveals the importance of such tools among societies living in cold environments. Yet, a different perspective is required to explain the high frequency of alternative activities observed such as tattooing, medical applications, basketry, etc. One potential link may follow from the ideas of cumulative culture, defined as "the modification, over multiple transmission episodes, of cultural traits (behavioral patterns transmitted through social learning) resulting in an increase in the complexity or efficiency of those traits" [33]. In other words, cumulative culture operates on the principle that cultural innovations are not reflections of a single individual but rather a result of building upon, modifying, and refining the technological and cultural achievements of previous generations. The gradual and directional accumulation of knowledge and skill leads to increasingly complex and efficient technologies and practices.

In the context of perforator tools, it is possible that the primary motivator behind needles and awls included the manufacture of clothing and other thermoregulation material culture items. However, as these tools became embedded within cultural traditions, the potential of needles and awls for alternative uses became apparent. Over successive generations, human populations could have observed, experimented, and taught others about the significance and multifaceted nature of needles and awls thus expanding the functional repertoire of perforators to include their use in ceremonies and rituals, medical suturing, basketry, tattooing, and other cultural uses beyond thermal technology. The diversification of needle and awl use not only highlights the practical functionality of the tools but also broader mechanisms of cumulative cultural evolution, in which the constant stream of exchange and engagement of ideas contributed to the versatility of tool traditions. Future research has the potential to further validate if cumulative culture was a primary contributor to a wide range of technological and social behaviors observed in relation to needles and awls in the ethnographic and archaeological records. Below, we consider several of the different activity types observed before concluding with additional implications for the archaeological record.

### Use types and temperature

As expected, perforator tools used to make several thermoregulation items like clothing, shoes, and snowshoes are used more commonly among colder environments. Surprisingly, there is no relationship between temperature and other thermoregulation items such as shelter or blankets. While sample size could be driving these outcomes, it is important to consider that shelter can offer shade to alleviate the effects of hot environments in addition to providing a climate-controlled space in colder landscapes [4]. Additionally, blankets can be other deeply rooted in sociocultural phenomena such as ceremonies, rituals, dances and rituals, horse blankets, social status, infant care, trade, and general comfort (see S1 Table) [34–38]. Blankets can also serve as a form of covering to provide shade and protect the skin from sun exposure, therefore alleviating the effects of warm stress.

Mats serve a variety of purposes including bedding, floormats, cushions for sitting, home décor, or used on the walls of housing structures. We anticipated that mats would be more prevalent in cold environments where they serve as a tool to reduce heat loss via thermal conditions with the ground. Consider, for example, backpackers who almost

invariably transport and sleep on ground pads. However, housing structures are applicable to maintaining thermoneutrality in both warm and cold environments, and nighttime temperatures may require this basic thermoregulatory technology even in otherwise warm climates. The elevated prevalence of mats in warmer environments may reflect comfort, privacy/soundproofing, and decoration these items have to offer in the general household rather than their provision of warmth. Additionally, warmer environments are related to increased frequency and intensity of disease, pathogens, and insect pests [39]. Therefore, floormats provide the opportunity for individuals to avoid pests and stay off the ground to maintain hygiene and health. Finally, floormats can provide a cooling effect associated with areas where heat can become trapped underground [40].

The non-significant relationship between temperature and basketry production reflects both the use of basketry in warm and cold climatic regions. The use of basketry production within warmer environments could be reflective of the availability of seeds, berries, and other plants collected for subsistence considering there is evidence of using basketry as a specialized food-processing technique (S1 Table) [41]. Furthermore, ethnographic evidence points to the creation and use of basketry in cold geographical regions in North America, with an example including the Yup'ik of Southwest Alaska and Natives of the Aleutian Islands using dried grasses to manufacture baskets [42]. This clearly demonstrates how basketry is not bound by certain climatic conditions but rather is pertinent to cultures across diverse ecologies.

The categories of ceremonies/rituals, storytelling, and medical care observed as part of this research are not correlated with temperature. This is not surprising considering the pancultural nature of such activities [43–45]. Tattooing and piercing are additional perforator tool activities with no correlation with temperature. While initially we expected that tattooing would be associated with warmer climates where skin tends to be more visible, there are many accounts of tattooing areas of body that are more visible to others even while clothing is worn such as the face, hands, and neck (S1 Table). It is also clear that tattoos have personal significance which helps to explain why needles are used to tattoo portions of the body that would be covered by clothing while exposed to cold external environments such as the arms, chest, shoulder, back, and legs (S1 Table). In other words, exposing large portions of the skin to external environmental conditions is not necessary for tattoos to carry significance, thus tattooing can occur in both warm and cold climates [46,47].

## Conclusions and implications for archaeological record

This study examined a hypothesized relationship between environmental temperature and perforator tool use in human societies by examining the prevalence of needle and awl technologies among 59 distinct North American Indigenous groups occupying variable climatic conditions. The data support the hypothesized increased dependence on perforator tools in cold temperatures. This research is one of the first to quantitatively estimate the effect of temperature on perforator tool use in the ethnographic record. The manufacture of clothing is an important motivator for the creation and use of needles and awls. While the data are consistent with this model, this study furthermore underscores the multifaceted utility of perforator tools across diverse cultural contexts.

As a proxy for interpreting perforator tools in the past, the findings presented here anticipate that the prevalence of Paleolithic bone needles and awls in archaeological contexts should increase in colder climates. These findings should be evaluated in the ethnographic records of other parts of the world. The data also imply that while clothing manufacture and maintenance is likely a major driver of perforator tool presence in the archaeological record, the presence of perforator tools does not guarantee their use as thermoregulatory technologies. Researchers should proceed with caution when evaluating needles and awls in archaeological contexts and avoid preconceived notions that these tools act as direct proxies for the manufacture of thermoregulatory material culture in the past. A more appropriate and tempered approach should acknowledge the multiplicity of behaviors associated with such tools. Nonetheless, a quantitative approach in which analysts consider not only the presence of perforator tools but also their abundance in the archaeological record should be used to strengthen inferences about thermoregulatory activities in a given region in the past. A future archaeological analysis of the relationship between temperature and perforator occurrence across the globe would certainly offer

a complimentary approach toward further clarifying the relationship between climatic conditions and perforator tool use in human societies. Additional studies may also explore the extent to which additional cultural variables could have impacted how and where needles and awls were used by past populations, with examples of such factors including mobility or sedentism, economic strategies, and patterns of landscape exploitation.

## Supporting information

**S1 Table. Excel spreadsheet featuring the raw data compiled for this study, based on texts available within the eHRAF World Cultures database.**
(CSV)

## Acknowledgments

We thank Allison Caine for providing insight into the ethnographic explanations for observed temperature relationships and perforator tool use. The authors also thank Bree Doering and Todd Surovell for reading earlier drafts of the paper. The ethnographic data analyzed for this project was provided by the eHRAF World Cultures database. Finally, this manuscript greatly benefits from comments provided by the editors and anonymous reviewers.

## Author contributions

**Conceptualization:** McKenna Lynn Litynski, Randall Haas.

**Data curation:** McKenna Lynn Litynski, Sean Field.

**Formal analysis:** McKenna Lynn Litynski, Sean Field, Randall Haas.

**Funding acquisition:** McKenna Lynn Litynski, Randall Haas.

**Investigation:** McKenna Lynn Litynski, Randall Haas.

**Methodology:** McKenna Lynn Litynski, Randall Haas.

**Project administration:** McKenna Lynn Litynski.

**Resources:** Sean Field.

**Software:** McKenna Lynn Litynski, Sean Field, Randall Haas.

**Supervision:** Sean Field, Randall Haas.

**Validation:** Randall Haas.

**Visualization:** McKenna Lynn Litynski, Randall Haas.

**Writing – original draft:** McKenna Lynn Litynski.

**Writing – review & editing:** McKenna Lynn Litynski, Sean Field.

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
