## [Decision Letter · Decision Letter 0]

20 Oct 2025

Dear Dr.  Litynski,

Thank you for submitting your manuscript to PLOS ONE. After careful consideration, we feel that it has merit but does not fully meet PLOS ONE’s publication criteria as it currently stands. Therefore, we invite you to submit a revised version of the manuscript that addresses the points raised during the review process.

We look forward to receiving your revised manuscript.

Kind regards,

Enza Elena Spinapolice, Ph.D

Academic Editor

PLOS ONE

Journal Requirements:

3. We note that Figure 1 in your submission contain map images which may be copyrighted. All PLOS content is published under the Creative Commons Attribution License (CC BY 4.0), which means that the manuscript, images, and Supporting Information files will be freely available online, and any third party is permitted to access, download, copy, distribute, and use these materials in any way, even commercially, with proper attribution. For these reasons, we cannot publish previously copyrighted maps or satellite images created using proprietary data, such as Google software (Google Maps, Street View, and Earth). For more information, see our copyright guidelines: http://journals.plos.org/plosone/s/licenses-and-copyright.

Reviewer's Responses to Questions

**Comments to the Author**

1. Is the manuscript technically sound, and do the data support the conclusions?

Reviewer #1: Yes

Reviewer #2: Yes

2. Has the statistical analysis been performed appropriately and rigorously?

Reviewer #1: I Don't Know

Reviewer #2: No

3. Have the authors made all data underlying the findings in their manuscript fully available?

Reviewer #1: Yes

Reviewer #2: Yes

4. Is the manuscript presented in an intelligible fashion and written in standard English?

Reviewer #1: Yes

Reviewer #2: Yes

Reviewer #1: Review of ‘’Ethnographic meta-analysis shows that thermoregulation activities predict needle and awl use in North America’’

This manuscript investigates the ethnographic uses of needles and awls in North America using the eHRAF World Cultures database. The authors test the hypothesis that thermoregulation was the primary driver of perforator use, while also considering alternative uses. Logistic regression analyses are applied to evaluate the relationship between minimum temperature of the coldest month (MTCM) and perforator use. The study finds that although non-thermoregulatory activities collectively dominate, clothing manufacture is the single most prevalent activity, and that colder climates are associated with higher probabilities of perforator use. The paper contributes to ongoing debates on the role of bone perforators in Paleolithic adaptations and offers a quantitative cross-cultural perspective.

The dataset and approach are innovative: the systematic use of eHRAF ethnographies, combined with climatic data, provides a large and well-documented sample that allows for robust cross-cultural analysis. The study is also framed around clear hypotheses and testable predictions, which are evaluated through clear statistical models. The paper demonstrates strong archaeological relevance by connecting ethnographic observations to broader interpretive challenges in Paleolithic archaeology, particularly concerning cold-climate adaptations. Finally, the authors’ commitment to transparency—through open data availability and plans to share code on GitHub—reflects best practices in open science and strengthens the credibility of the study.

I should note, however, that as a reviewer I do not have the expertise to verify the reliability of the R code provided by the authors, nor to assess whether more suitable climatic datasets might be available in the literature for a more robust analysis. Further clarification on why MTCM was prioritized over annual mean temperature or effective temperature would help.

The dichotomy between “thermoregulatory” and “alternative” uses appears somewhat rigid. This is particularly evident in the case of mats, for which the analysis revealed no correlation with cold environments. While the authors emphasize that this result runs counter to theoretical expectations, it more likely highlights the influence of additional variables, such as mobility. For example, communities inhabiting colder regions may have been more mobile than those in warmer environments, which could have limited the transport and use of mats depending on the degree of mobility or sedentism. The paper would benefit from further consideration of such parameters, including mobility, economic strategies, and patterns of landscape exploitation, as potential factors shaping perforator use.

The discussion introduces cumulative culture as a potential explanatory lens but leaves this underdeveloped. Expanding this theoretical point would be useful.

Overall, the paper is interesting, appropriately written and well organized, and I believe it merits publication pending minor revisions.

Reviewer #2: This is a really interesting manuscript, which I enjoyed reading a great deal. Neat idea and interesting findings.

Besides some of the minor points flagged up in the attached annotated pdf, my main worries relate to the statistical analysis.

1) The WorldClim data are present-day data, so they do NOT reflect the climatic conditions during the time of observation (which is recorded in HRAF) nor conditions during which needle-work evolved in the societies under study.

2) There is very likely a great deal of historical relatedness and hence statistical non-independence in the sample used - this is very likely to have serious implications for the statistical power of the analysis (e.g. actual sample size) and potentially the results, all of which remains unaccounted for at present.

The authors allude to the limitations of the ethnographic perspective but do so in a very unspecific (and non-analytical) way.

I would also like to see the authors explore and report similar analyses for other temp-related variables - pls see my comment on this in the pdf.

Showing a figure of some of the cool awls and needles from across these societies would also be really nice and relevant.

Some additional comments, ideas and relevant references are in the pdf.

I look forward to seeing this study published in due time.

**Do you want your identity to be public for this peer review?** For information about this choice, including consent withdrawal, please see our Privacy Policy

Reviewer #1: No

Reviewer #2: **Yes:** Felix Riede

---

## [Author Response · Author response to Decision Letter 1]

27 Dec 2025

Reviewer/Editor Comments and Responses

PONE-D-25-39223

Ethnographic meta-analysis shows that thermoregulation activities predict needle and awl use in North America

PLOS ONE

Comment: Please ensure that your manuscript meets PLOS ONE's style requirements, including those for file naming.

Response: We have reformatted the manuscript such that it meets PLoS ONE's style requirements based on the PDF documents shared. This includes changing figure references in the text, altering the referencing cited so that it meets the required numbering system, changing the font type and size for headings and sub-headings, etc.

Comment: When completing the data availability statement of the submission form, you indicated that you will make your data available on acceptance. We strongly recommend all authors decide on a data sharing plan before acceptance, as the process can be lengthy and hold up publication timelines. Please note that, though access restrictions are acceptable now, your entire data will need to be made freely accessible if your manuscript is accepted for publication. This policy applies to all data except where public deposition would breach compliance with the protocol approved by your research ethics board. If you are unable to adhere to our open data policy, please kindly revise your statement to explain your reasoning and we will seek the editor's input on an exemption. Please be assured that, once you have provided your new statement, the assessment of your exemption will not hold up the peer review process.

Response: The R code (now accounting for spatial autocorrelation) associated with the statistical models is now publicly available through GitHub and mentioned as S1 Text in the Supporting Information section. All other associated raw data from the ethnographic meta-analysis is associated with the S1 Table.

Comment: We note that Figure 1 in your submission contain map images which may be copyrighted. All PLOS content is published under the Creative Commons Attribution License (CC BY 4.0), which means that the manuscript, images, and Supporting Information files will be freely available online, and any third party is permitted to access, download, copy, distribute, and use these materials in any way, even commercially, with proper attribution. For these reasons, we cannot publish previously copyrighted maps or satellite images created using proprietary data, such as Google software (Google Maps, Street View, and Earth).

Response: We understand that PLoS ONE cannot publish previously copyrighted maps or satellite images created using proprietary data. The data that we used to create the heat map (now noted as Figure 2) include: (1) Points based on information about each ethnographic group's geographical centroid associated with the eHRAF World Cultures database. (2) Temperature data associated with the open-access WorldClim database. Considering all of the corresponding data are publicly available, we believe that the presentation of the heat map in PLoS One will not violate any copyright policies.

Comment: Please include captions for your Supporting Information files at the end of your manuscript, and update any in-text citations to match accordingly.

Response: We revised accordingly and included captions for supporting information (S1 Table and S1 Text) at the end of our manuscript. We updated any in-text citations to match this accordingly.

Comment: The dichotomy between “thermoregulatory” and “alternative” uses appears somewhat rigid. This is particularly evident in the case of mats, for which the analysis revealed no correlation with cold environments. While the authors emphasize that this result runs counter to theoretical expectations, it more likely highlights the influence of additional variables, such as mobility. For example, communities inhabiting colder regions may have been more mobile than those in warmer environments, which could have limited the transport and use of mats depending on the degree of mobility or sedentism. The paper would benefit from further consideration of such parameters, including mobility, economic strategies, and patterns of landscape exploitation, as potential factors shaping perforator use.

Response: The reviewer makes an excellent point in that there are certainly many different variables that could have influenced how and where needles and awls were used by past human populations. However, we believe that addressing all the proposed variables is outside the scope of this research and could be addressed in separate paper(s). Nonetheless, we address these valid concerns by including a sentence in the concluding paragraph that states, "Additional studies may also explore the extent to which additional cultural variables could have impacted how and where needles and awls were used by past populations, with examples of such factors including mobility or sedentism, economic strategies, and patterns of landscape exploitation."

Comment: The discussion introduces cumulative culture as a potential explanatory lens but leaves this underdeveloped. Expanding this theoretical point would be useful.

Response: We have expanded upon explanations of cumulative culture here:

The finding of a clear relationship between temperature and perforator use in the ethnographic record reveals the importance of such tools among societies living in cold environments. Yet, a different perspective is required to explain the high frequency of alternative activities observed such as tattooing, medical applications, basketry, etc. One potential link may follow from the ideas of cumulative culture, defined as “the modification, over multiple transmission episodes, of cultural traits (behavioral patterns transmitted through social learning) resulting in an increase in the complexity or efficiency of those traits” [28]. In other words, cumulative culture operates on the principle that cultural innovations are not reflections of a single individual but rather a result of building upon, modifying, and refining the technological and cultural achievements of previous generations. The gradual and directional accumulation of knowledge and skill leads to increasingly complex and efficient technologies and practices. In the context of perforator tools, it is possible that the primary motivator behind needles and awls included the manufacture of clothing and other thermoregulation material culture items. However, as these tools became embedded within cultural traditions, the potential of needles and awls for alternative uses became apparent. Over successive generations, human populations could have observed, experimented, and taught others about the significance and multifaceted nature of needles and awls thus expanding the functional repertoire of perforators to include their use in ceremonies and rituals, medical suturing, basketry, tattooing, and other cultural uses beyond thermal technology. The diversification of needle and awl use not only highlights the practical functionality of the tools but also broader mechanisms of cumulative cultural evolution, in which the constant stream of exchange and engagement of ideas contributed to the versatility of tool traditions. Future research has the potential to further validate if cumulative culture theory was a primary contributor to a wide range of technological and social behaviors observed in relation to needles and awls in the ethnographic and archaeological records.

Comment: The WorldClim data are present-day data, so they do NOT reflect the climatic conditions during the time of observation (which is recorded in HRAF) nor conditions during which needle-work evolved in the societies under study.

Response: For this analysis, we use WorldClim version 2.1 and Bioclimatic variable 6. Based on the WorldClim website, the WorldClim Bioclimatic variables for WorldClim are the average for the years 1970-2000. Most of the ethnographic data from eHRAF was collected between the early 1800s up until 2000. The reviewer raises an important point that is perennial to archaeological research. The climates of the present rarely reflect the climates of the past. However, ours is an ethnographic analysis, and although the ethnographic cases are from the past, the temporal mismatches are on the order of 0-150 years. At this temporal scale and given that our dataset spans some 50 degrees C, any MTCM errors are certain to be trivial. Although we could use paleoclimatic data, such datasets, which are based on a series of interpolations and assumption, would likely introduce much greater error than WorldClim. We therefore feel confident that WorldClim is the most appropriate dataset for the analysis and most likely to minimize error in this variable. We have added text to clarify this decision.

Comment: There is very likely a great deal of historical relatedness and hence statistical non-independence in the sample used - this is very likely to have serious implications for the statistical power of the analysis (e.g. actual sample size) and potentially the results, all of which remains unaccounted for at present.

Response: We greatly appreciate this feedback and recognize the importance of accounting for the longstanding issue of spatial autocorrelation within our statistical models evaluating the relationship between temperature and ethnographic perforator observations. We have followed the reviewer’s suggestion. To account for autocorrelations (phylogenetic and spatial), we reformulated our regressions as a series of linear mixed-effect models that include latitude and longitude coordinates interactive random effects. Autocorrelation is modeled using a Matérn function as implemented by the spaMM R package. This function effectively controls for autocorrelation and therefore accounts for statistical non-independence in our data. The updated analyses reveals the same basic patterns with some important refinements. First, we observe that MTCM induces the anticipated negative relationships with the probabilities of observing thermoregulatory technologies. That said, as the reviewer anticipated, accounting for autocorrelation effects did effectively diminish the sample size and the statistical power of the results. Nonetheless, the trends remain consistent with theory across the various thermoregulatory subsets and pass statistical muster even if at higher thresholds of acceptance. We do not wish to devolve into p-hacking debates, and we underscore that the use of p-values in mixed-effect models (e.g. autocorrelation modeling) is a matter of debate. We feel that the end result, which uses all available data and accounts for autocorrelation, offers the most honest answers to the question. Second, we observe that the previously identified positive relationship between MTCM and observation of mat and basketry use now fades away with the consideration of autocorrelation. The results also account for significant spatial autocorrelation bias regarding the “basketry” and “mats”. While the initial generalized linear models indicated a significant positive relationship between temperature and the use of perforator tools for these specific activities, the results of the updated linear mixed-effect models that account for spatial autocorrelation reveal no relationship between temperature and the use of perforators for the manufacture and use of basketry and mats. We are grateful for the Reviewers suggestion, which has helped us to avoid what we previously perceived to be a surprising false-positive that was not anticipated by our theory. Overall, we are pleased with these new results and are more confident in the conclusion that environmental temperatures do influence the use of needles and awls. Again, we thank the reviewer for this prescient insight.

Comment: The authors allude to the limitations of the ethnographic perspective but do so in a very unspecific (and non-analytical) way.

Response: We change the paragraph on ethnographic limitations to the following:

We remain alert to several limitations and biases that come with deriving data from eHRAF World Cultures. First, we recognize the impact of ethnocentrism and how certain activities associated with needles and awls may not be included in the ethnographic documents. In this light, we also recognize the inherent gender bias that can skew the perspectives, experiences, and information shared in the ethnographic record [21-23]. We do not make speculations about the number of missing activity types as a function of ethnocentrism or gender bias. Rather we focus on qualitatively and quantitatively evaluating the ethnographic data and associated perforator activities directly recorded in eHRAF World Cultures. Additionally, we assess the relationship between perforator tool activity types and temperature by qualitatively evaluating geographical data by plotting our ethnographic database in space in addition to selecting specific search terms specific to perforator tools. The coded data and the temperature trends associated with perforator tool use remains relevant and useful to understanding the wide variety of uses for needles and awls.

Comment: Showing a figure of some of the cool awls and needles from across these societies would also be really nice and relevant.

Response: We have added a new figure (Fig. 1) featuring illustrations of bone needles and awls for visual purposes.

Comment: I would also like to see the authors explore and report similar analyses for other temp-related variables - pls see my comment on this in the pdf. Comment in PDF: Why only MTCM - other variables could be explored, also because some cold stress can be mitigated more readily by physiological adaptations and some environments afford thermoregulation using pyro-technology. Ordonez & Riede used a 'limiting factor' approach to consider which climatic variables constrained human pop densities in the European Late Palaeolithic/Early Mesolithic. Across this (mostly very cold) period, there is a marked increase and subsequent decline in needle occurrence and this, they hypothesise (narratively) might relate to not just temp or MTCM but acutely changing pressures of different forms of cold.

Response: The reviewer raises an important point that we did initially consider. Our rationale for the use of MTCM, which we have added to the text, is because MTCM most precisely captures the selective force driving thermoregulatory behaviors. Other scholars have noted that it is often the extreme events that shape selection—not the averages (Moran 2008;Grant et al. 2017. Thus this variable is theoretically motivated compared to other climatic variables, which would not be so theoretically justified. Other temp related variables, such as effective temp or avg temp, would furthermore be strongly correlated with MTCM and would almost certainly yield similar results. Nonetheless, the lack of theoretical justification and consideration of other variable would quickly devolve to a statistical fishing expedition, which we prefer to avoid.

Comment: "To test the prediction that the majority of perforator tools were used for thermoregulation, the total number of ethnographic documents within the “thermoregulation” and “alternative” categories were converted to percentages with the exception that the proportion of thermoregulatory activities should exceed that of other activities." I don't understand this sentence. Should exception read expectation?

Response: Yes, the word "exception" in this sentence should indeed read “expectation”. This has been corrected in the manuscript.

Comment: Two-hundred and twenty-three mentions of needles and/or awls in the eHRAF ethnographic record fall under the generic “Toolkit” category and thus cannot be assigned to a specific activity." Do these nonetheless show geographic (i.e. climatic) structure?

Response: These needles and awls belonging to this generic "Toolkit" category are included in statistical models considering the relationship between temperature data and the presence of needles and awls in the ethnographic record. To make this clearer, we add the following sentence: "Tools that fall under t

---

## [Decision Letter · Decision Letter 1]

13 Feb 2026

Ethnographic meta-analysis shows that thermoregulation activities predict needle and awl use in North America

PONE-D-25-39223R1

Dear Dr. Litynski,,

We’re pleased to inform you that your manuscript has been judged scientifically suitable for publication and will be formally accepted for publication once it meets all outstanding technical requirements.

Kind regards,

Enza Elena Spinapolice, Ph.D

Academic Editor

PLOS One

Additional Editor Comments (optional):

Reviewers' comments:

Reviewer's Responses to Questions

**Comments to the Author**

Reviewer #1: All comments have been addressed

Reviewer #2: All comments have been addressed

2. Is the manuscript technically sound, and do the data support the conclusions?

Reviewer #1: Yes

Reviewer #2: Yes

3. Has the statistical analysis been performed appropriately and rigorously?

Reviewer #1: Yes

Reviewer #2: Yes

4. Have the authors made all data underlying the findings in their manuscript fully available?

Reviewer #1: Yes

Reviewer #2: Yes

5. Is the manuscript presented in an intelligible fashion and written in standard English?

Reviewer #1: Yes

Reviewer #2: Yes

Reviewer #1: Manuscript: Ethnographic meta-analysis shows that thermoregulation activities predict needle and awl use in North America

I have reviewed the revised manuscript and the authors’ responses to prior comments. The manuscript has improved substantially.

The authors have appropriately addressed concerns regarding statistical non-independence by incorporating spatial autocorrelation into their linear mixed-effect models. This significantly strengthens the analytical framework and increases confidence in the reported relationships between temperature (MTCM) and perforator use. The clarification of the temperature proxy, the expanded discussion of cumulative culture, the improved articulation of ethnographic limitations, and the reorganization of Table 1 all enhance the clarity and rigor of the study.

The revised analyses continue to support the central conclusions: thermoregulatory activities are strongly associated with perforator use, and colder temperatures predict a higher likelihood of perforator occurrence, while alternative uses show no consistent temperature relationship. I have no further substantive concerns.

Recommendation: Accept without modification.

Reviewer #2: Thanks for addressing my comments conscientiously. Regarding my remark on the status of your paper as a meta-analysis, please be advised that - just from my own experience as PLoS author - you'll get grief from the editorial office. PLoS has this very formal policy that unless you followed formal meta-analysis guidelines (i.e. PRISMA) you cannot call it that. So I do suggest you change the title and use 'quantitative cross-cultural analysis' instead of meta-analysis, for example. Otherwise, cool study, congrats.

**Do you want your identity to be public for this peer review?** For information about this choice, including consent withdrawal, please see our Privacy Policy

Reviewer #1: No

Reviewer #2: **Yes:** Felix Riede

---

## [Editor Report · Acceptance letter]

PONE-D-25-39223R1

PLOS One

Dear Dr. Litynski,

I'm pleased to inform you that your manuscript has been deemed suitable for publication in PLOS One. Congratulations! Your manuscript is now being handed over to our production team.

Kind regards,

on behalf of

Dr. Enza Elena Spinapolice

Academic Editor

PLOS One